# Antenatal Assessment of the Prognosis of Congenital Diaphragmatic Hernia: Ethical Considerations and Impact for the Management

**DOI:** 10.3390/healthcare10081433

**Published:** 2022-07-30

**Authors:** Kévin Le Duc, Sébastien Mur, Dyuti Sharma, Rony Sfeir, Pascal Vaast, Mohamed Riadh Boukhris, Alexandra Benachi, Laurent Storme

**Affiliations:** 1Department of Neonatology, Pôle Femme-Mère-Nouveau-Né, Hôpital Jeanne de Flandre, Centre Hospitalier Universitaire de Lille, F-59000 Lille, France; sebastien.mur@chu-lille.fr (S.M.); dyuti.sharma@chu-lille.fr (D.S.); rony.sfeir@chu-lille.fr (R.S.); riadh.boukhris@chu-lille.fr (M.R.B.); laurent.storme@chu-lille.fr (L.S.); 2ULR 2694-METRICS: Évaluation des Technologies de Santé et des Pratiques Médicales, Axe Environnement Périnatal et Santé, Centre Hospitalier Universitaire de Lille, F-59000 Lille, France; 3Center for Rare Disease Congenital Diaphragmatic Hernia, Jeanne de Flandre Hospital, Centre Hospitalier Universitaire de Lille, F-59000 Lille, France; pascal.vaast@chu-lille.fr; 4Department of Obstetrics, Jeanne de Flandre Hospital, Centre Hospitalier Universitaire de Lille, F-59000 Lille, France; 5Department of Obstetrics, Hôpital Antoine Béclère, Assistance Publique-Hôpitaux de Paris, F-92140 Clamart, France; alexandra.benachi@aphp.fr

**Keywords:** congenital diaphragmatic hernia, morbidity, prenatal prognostic, self-fulfilling prophecies

## Abstract

Congenital diaphragmatic hernia (CDH) is associated with abnormal pulmonary development, which is responsible for pulmonary hypoplasia with structural and functional abnormalities in pulmonary circulation, leading to the failure of the cardiorespiratory adaptation at birth. Despite improvement in treatment options and advances in neonatal care, mortality remains high, at close to 15 to 30%. Several risk factors of mortality and morbidities have been validated in fetuses with CDH. Antenatal assessment of lung volume is a reliable way to predict the severity of CDH. The two most commonly used measurements are the observed/expected lung to head ratio (LHRo/e) and the total pulmonary volume (TPV) on MRI. The estimation of total pulmonary volume (TPVo/e) by means of prenatal MRI remains the gold standard. In addition to LHR and TPV measurements, the position of the liver (up, in the thorax or down, in the abdomen) also plays a role in the prognostic evaluation. This prenatal prognostic evaluation can be used to select fetuses for antenatal surgery, consisting of fetoscopic endoluminal tracheal occlusion (FETO). The antenatal criteria of severe CDH with an ascended liver (LHRo/e or TPVo/e < 25%) are undoubtedly associated with a high risk of death or significant morbidity. However, despite the possibility of estimating the risk in antenatal care, it is difficult to determine what is in the child’s best interest, as there still are many uncertainties: (1) uncertainty about individual short-term prognosis; (2) uncertainty about long-term prognosis; and (3) uncertainty about the subsequent quality of life, especially when it is known that, with a similar degree of disability, a child’s quality of life varies from poor to good depending on multiple factors, including family support. Nevertheless, as the LHR decreases, the foreseeable “burden” becomes increasingly significant, and the expected benefit is increasingly unlikely. The legal and moral principle of the proportionality of medical procedures, as well as the prohibition of “unreasonable obstinacy” in all investigations or treatments undertaken, is necessary in these situations. However, the scientific and rational basis for assessing the long-term individual prognosis is limited to statistical data that do not adequately reflect individual risk. The risk of self-fulfilling prophecies should be kept in mind. The information given to parents must take this uncertainty into account when deciding on the treatment plan after birth.

## 1. Introduction

Congenital diaphragmatic hernia (CDH) is generally associated with the ascension of the abdominal viscera towards the thorax, and abnormalities in pulmonary development are responsible for pulmonary hypoplasia and structural and functional abnormalities in pulmonary circulation. Its functional consequences are particularly heterogeneous, since some infants are asymptomatic at birth, while others present major failure of cardiorespiratory adaptation to extrauterine life [1].

The pathophysiology of this malformation is complex, and despite progress in intensive care, neonatal mortality remains high, close to 15 to 30%, mainly due to pulmonary hypoplasia and pulmonary arterial hypertension (PAH) [2]. About 10% of children with CDH die, from the neonatal period to the first years of life, from respiratory and/or digestive complications [3]. Additionally, CDH is associated with high morbidity, which affects about half of surviving infants. The main sequelae observed are respiratory (chronic PAH, bronchopulmonary dysplasia, susceptibility to viral infections), digestive (gastroesophageal reflux, oral aversion), and orthopedic (scoliosis) [3]. 

Several risk factors of mortality and morbidities have been validated in fetuses with CDH [4,5]. They suggest that an improved understanding of the pathophysiology of the condition is needed to improve the care of these children. The objectives of care during the neonatal period are to reduce immediate mortality, mainly linked to a failure to adapt to extrauterine life, but also to implement, from birth, measures to prevent long-term morbidity. In 2008, the Rare Disease Reference Center: Congenital Diaphragmatic Hernia was created in France. One of the center’s missions is to offer a care pathway for the care of fetuses and children with CHD. This incorporates the antenatal estimate of CDH severity and threat to life.

## 2. Antenatal Life Threat Assessment

The prognosis of children with CDH depends on multiple factors: (1) isolated or syndromic form; (2) gestational age; (3) on which side the CDH is located; and (4) the severity of pulmonary hypoplasia. CDH is usually sporadic, although rare inherited forms have been reported. In 30% of cases, CDHs are associated with other malformations and/or chromosomal abnormalities [6]. The prognosis of CDH is then essentially determined by the prognosis of the associated syndrome. Straight diaphragmatic hernias account for 15% of CDH cases diagnosed prenatally. The overall severity of right versus left CDH is controversial. In the case series by Thomas Schaible et al., the mortality rate of the two types of CDH is the same, but the long-term lung morbidity rate is higher for right CDH [7]. Antenatal prognostic evaluation of right CDH is more difficult than for left CDH [8]. The liver is ascended in right CDHs, and the amount by which the liver is ascended has not been evaluated as a prognostic factor. 

Premature birth is also recognized as a factor determining the risk of death. Of the 5022 children in the CDH International Register, 3895 were born at term (78%) and 1127 were born prematurely (22%) [9]. Overall survival was 68.7%. Preterm neonates had a higher percentage of chromosomal abnormalities (4% term vs. 8% premature) and major cardiac abnormalities (6% term vs. 11% premature). The overall survival of premature infants is lower than in full-term infants (75% term vs. 55% premature). In 2010, survival decreased with gestational age, reaching 35% in children with a GA under 30 weeks [9]. 

Pulmonary hypoplasia and pulmonary arterial hypertension (PAH) are the two major determinants of neonatal mortality and morbidity. Antenatal assessment of lung volume is a reliable way to predict the severity of CDH. Antenatal prognostic evaluation is important because it allows (1) the comparison of care between institutions and (2) the selection of fetuses who can benefit from the placement of a tracheal balloon by fetoscopy. Pulmonary volume assessment is now possible. In practice, the two most commonly used measurements are the observed/expected lung to head ratio (LHR o/e) and that of the total pulmonary volume (TPV) on MRI [1]. The LHR o/e measurement is routinely used. This measurement allows for an indirect evaluation of the contralateral pulmonary volume and therefore of pulmonary hypoplasia. The sensitivity of this test in the prediction of survival is only 46% with a 10% rate of false positives (regardless of the position of the liver). However, it is a reproducible measure in a trained team, independent of gestational age [5]. The estimation of total pulmonary volume (TPVo/e) by prenatal MRI remains the gold standard [10]. Both lungs, including the herniated ipsilateral lung, can be measured, while this is rarely possible by means of ultrasound [11]. However, this examination is not available everywhere on an urgent basis. The LHR o/e remains the first examination to be carried out, later supplemented by the MRI. In addition to LHR and TPV measurements, the position of the liver plays a role in the prognostic evaluation. MRI has allowed for more accurate volumetric estimation of the ascended portion of the liver and has improved the prediction of survival [12]. The assessment of the position of the liver has been simplified by classifying the position of the stomach on the LHR section. The closer the stomach is to the atrioventricular valves, the more the liver is ascended [13]. 

Recently, it has been demonstrated that the gestational age at which CDH is diagnosed is an independent predictor of postnatal prognosis: early diagnosis is associated with a higher mortality rate [14].

The limiting factor for estimating the prognosis of children with CDH is the prediction of PAH. This estimate is not well-correlated with the severity of PAH [15]. Studies have attempted to correlate prenatal pulmonary vascularization with postnatal PAH, including attempting to visualize and quantify pulmonary vascularization and blood flow. Pulmonary artery Doppler with resistance index (RI), pulsatility index (PI), and systolic peak velocity (SPV) measurements are not used in practice, as they have been shown to be dependent on LHRo/e [16,17]. Recent data on PAH mechanisms in CDHs explain why antenatal estimation is not predictive of PAH levels. Indeed, in most cases, PAH is of post-capillary origin related to left heart dysfunction, at least in the first few days of life [18,19]. Moreover, postnatal definition of PH has some limitations. It is also important to note at which postnatal time points PH assessment is performed. Mechanisms of PH change postnatally and usually PH decreases within the first 1–3 weeks.

## 3. Relevance of the Antenatal Life-Threat Assessment in Relation to Management

This analysis of the literature clearly indicates that it is possible to estimate the risk of death for a fetus with CDH. There is consensus among obstetricians regarding the methods of this evaluation. The measures have been standardized and are the subject of regular training which allows for a good reliability of the evaluation. 

### 3.1. Impact on Antenatal Care

This prenatal prognostic evaluation can therefore be used to select fetuses for antenatal surgery. This consists of the placement of a tracheal balloon by means of fetoscopy. 

During fetal life, the alveolar epithelium secretes fluid, which plays a crucial role in the development and growth of the lung. In the experimental model of a sheep fetus with CDH, tracheal ligation at least partially prevents pulmonary hypoplasia [20,21]. Lifting the occlusion before birth makes it possible to improve the secretion of the surfactant by the type II pneumocytes. In 2001, Deprest and Nicolaides performed the first tracheal occlusion in a human fetus using a fetoscopy technique to place a detachable balloon under the vocal cords [22]. The balloon should ideally be removed by fetoscopy before birth.

The first observational studies suggest that the placement of an intra-tracheal balloon placed around the 28th week improves the survival of children with a severe form of CDH [23,24]. Only one randomized study has been published by the R. Ruano team for severe forms, which concluded that there was an increase in the survival of cases in which FETO was performed, while the survival rate in the control group was very low [25]. Nevertheless, fetoscopy exposes the fetus to an increased risk of prematurity. Premature membrane rupture represents the main complication of this technique. 

Tracheal lesions were reported early in the use of the technique when the balloons were placed early on in pregnancy. Fayoux et al. performed tracheal endoscopies on seven children who had undergone a FETO [26]. Segmental tracheomegaly is observed with complete tracheal collapse at the end of the expiratory phase followed by progressive distension of the trachea during the inspiratory phase. Long-term studies are needed, but since FETOs are no longer performed before 28 weeks, it appears that the rate of tracheal complications is low. FETO increased the risks of preterm, prelabor rupture of membranes and preterm birth [27].

Two randomized trials were published recently to assess the benefits and risks of this technique. A randomized trial enrolled fetuses with “moderate” CDH, i.e., an LHR o/e between 25 and 34.9% and an intrathoracic or non-intrathoracic liver and LHR o/e between 35 and 44.9% with an ascended liver. The placement of the balloon is carried out between 30 + 0 and 31 + 6 weeks of amenorrhea and the withdrawal is performed at 34 and 34 + 6 weeks of amenorrhea. The primary objective of this trial is to compare the treatment by tracheal occlusion with a standard postnatal treatment in terms of the occurrence of pulmonary bronchodysplasia of infant survival to discharge and survival without oxygen supplementation at 6 months of age. This trial did not show a significant benefit of FETO [27]. The other trial enrolled fetuses with severe CDH whose LHRo/e is <25% with an ascended or non-ascended liver. The placement is carried out between 28 + 0 and 29 + 6 weeks of amenorrhea and the removal is performed at 34 and 34 + 6 weeks of amenorrhea. FETO resulted in a significant benefit over expectant care with respect to survival to discharge, and this benefit was sustained to 6 months of age [28].

### 3.2. Impact on Postanal Care

Children with a form of CDH that is considered severe based on antenatal criteria are undoubtedly at high risk of death or subsequent significant morbidity. To ensure the possible survival of these children, an extended stay in a resuscitation unit/intensive care and the implementation of intensive treatments such as ECMO may be necessary. Medical and surgical care is intensive and complex, requiring a massive investment by teams and parents, but also by society, which has to bear the financial cost of treatment and possible complications. Whatever the quality of care and family and social support, there is a “burden” that the child and their parents must bear. For the child, in the short term, it is a matter of coping with repeated physical pain, even if most of it can be prevented by appropriate treatment. In the most severe cases, multiple hospitalizations for respiratory decompensation or nutritional difficulties may be necessary during the first few years of life. For the parents, despite the support of the care team, the immediate burden to bear is that of low morale, worry, anxiety, family or professional difficulties related to hospitalization, or even mourning if the child dies; in the longer term, the burden can be from problems related to residual pathologies. In other words, the burden to be borne, sometimes onerous and lasting, can have consequences on family and professional life. All of these “burdens” justify the legitimate questioning of limits to be set in the best interests of the child. Should ante- and post-natal care for children with severe CDH be considered unreasonable? Is it not disproportionate? 

However, despite the possibility of estimating the risk in antenatal care, it is difficult to determine what is in the child’s best interest, as there are many uncertainties:(1)Uncertainty about individual short-term prognosis. CDH is a particularly heterogeneous malformation. Statistical data on a population poorly reflect the individual reality. Most studies to assess the prognostic value of the pulmonary volume estimate are based on a retrospective analysis of the results of several different teams. However, despite a relative standardization of care, mortality varies significantly from one team to another after adjustment for pulmonary volume measurements [29]. Moreover, the possibility of a self-fulfilling prophecy cannot be ruled out. This phenomenon has been described in the context of intensive care in situations where treatment is limited or discontinued [30]. In patients who are predicted to be at high risk of death despite continued treatment, the team may opt to discontinue treatment. In this case, mortality is high, regardless of the original value of the predictive criterion. The sustainability of this approach is reinforced because the mortality rate in this population is then high. This phenomenon has been described in particular in premature infants, or in adults who have experienced a stroke [31]. In this case, it is the prediction itself that contributes to increased mortality. Thus, the individual prediction of respiratory difficulties at birth and long-term outcomes are uncertain, even if, statistically, risk factors have been validated;(2)Uncertainty about long-term prognosis. While respiratory and nutritional morbidity can be significant in the first few months of life, the majority of difficulties recede during the early years of life. It is rare for a child with CDH to have disabling long-term effects or a major disability [3]. Thus, the term “sequelae” of CDH appears inappropriate in the majority of cases, as the sometimes major difficulties improve in the early years of life;(3)Uncertainty about subsequent quality of life, especially when it is known that, with a similar degree of disability, a child’s quality of life varies from poor to good depending on multiple factors, including the family environment. Indeed, the way in which parents accept their child’s difficulties determines his or her subsequent quality of life [32,33,34]. It appears that a good quality of life is difficult to define, and that its appreciation is subjective by nature. At what threshold is it decided that the difficulties or possibly the consequences, or even the quality of life, no longer justify curative care [35]? Only the child and their parents and relatives can evaluate the “burdens” they have borne and the benefits obtained, and this only after a clinical course that is difficult to define temporally, ranging from a few months to a few years. Thus, the definition may vary between parents, caregivers or society [33,34]. In this context, claiming that caregivers hold objective knowledge and that parents have only a subjective view of reality is a fantasy of omnipotence [33,34,35,36,37]. An interesting study confirmed this hypothesis in the case of CDH. The quality of life of adults with CDH at birth was compared with that of a healthy control group. Despite a morbidity similar to that described in the literature, including digestive, nutritional and respiratory conditions, the average quality of life of adults with CDH, estimated by the patients themselves, was not significantly different from those in the control group [38].

Nevertheless, as the LHR decreases, the foreseeable “burden” becomes increasingly significant, and the expected benefit is increasingly unlikely. The legal and moral principle of proportionality of medical procedures, as well as the prohibition of “unreasonable obstinacy” in all investigations or treatments undertaken, is necessary in these situations. Finally, it should be noted that death, possibly resulting from a decision to medically terminate the pregnancy, to stop resuscitative care and to resort to palliative care, does not remove the “burden” to be borne, which then corresponds to the events preceding death and then to the suffering related to the loss of the child. Parents must be the preferred interlocutors to determine the boundary between what seems to them to be a life of acceptable quality or not. These decisions are tied to their own destiny. In addition to being the legal representatives charged with overseeing the best interests of their children, the parents’ perception of the situation takes into account the specificities of their histories and personalities. This subjective perception must be seen as one of the essential aspects of the data to be integrated into the decision-making process. The prerequisite for parental involvement in decision-making processes is clear and fair information.

However, the scientific and rational basis for judging the long-term individual prognosis is limited to statistical data that do not adequately reflect individual risk. Under these conditions, there is the risk that the content of the information will differ according to the person’s own subjective perception of the risk involved. It will therefore be necessary to pay particular attention to the fact that the content of the information delivered is the result of a reflection of the healthcare team and not a single assessment by one person. This is one of the necessary conditions for information to be truly fair. The corollary of this argument is that a multidisciplinary meeting should take place regarding any parental information, as soon as any change to the treatment plan is considered: indeed, this initial information has decisive weight on the parental position. The greater the parents’ contribution to the decisions, the greater the importance of the meeting to cooperatively develop information that is a faithful reflection of the consensus and is acceptable to all.

## 4. Conclusions

Threat-to-life estimates for a fetus with CDH can be formulated by measuring lung volume by MRI or by measuring LHR. The values are well-correlated with the risk of pulmonary hypoplasia. In antenatal care, they make it possible to select at-risk fetuses that can benefit from the placement of a tracheal balloon. In postnatal settings, however, the scientific and rational basis for judging individual prognoses is limited to statistical data that do not adequately reflect individual risk. The risk of self-fulfilling prophecies should be kept in mind. The information given to parents must take this uncertainty into account when deciding on the treatment plan after birth. 

## Data Availability

Not applicable.

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
