# Peer review of "Antenatal Assessment of the Prognosis of Congenital Diaphragmatic Hernia: Ethical Considerations and Impact for the Management"

_healthcare, 2022, doi:10.3390/healthcare10081433_

Round 1

Reviewer 1 Report

In this current review, Le Duc et al describe the link between antenatal assessment and the postnatal course in patients with CDH. 

This review describes well the underlying pathological changes of CDH as well as the fetal assessment and potential therapeutic options. The study raises important issues in the care of CDH patients: the individual burden, the burden for parents and caregivers and the burden for the health system. These are questions to be asked and answered for every patient in an individual manner. The difficulty to predict the course of a patient based on the prenatal data is well described. Also the ethical dilemma physicians have to deal with during counseling and decision making are thoroughly discussed. In this perspective it is a great paper very worth reading.

Major change:

I would recommend to stress out in the title that the manuscript deals with the the ethical implications of antenatal findings. The manuscript is unique in this point and this should not get lost.

Impact on antenatal care:

Lines 154 following: Please update this section with the now published results from the moderate and the severe arm of the TOTAL trial.

Minor changes: 

Lines 121 following: The postnatal definition of PH has some limitations. It is also important at which timepoints postnatally PH assessment is performed. Postnatally PH and PVR have a certain dynamic and does decrease within the first 1-3 weeks. Please include this limitation.

Lines 125/126: better to write risk of death than risk of life.

Line 207 change HDC to CDH

Line 214: Change good life to good quality of life.

Reviewer 2 Report

First of all, I want to note that it has been a pleasure review your manuscript. I think this is an interesting topic for clinicians who manage this  dangerous malformation with high mortality and associated morbidity.

The type of manuscript is a commentary on the impact of antenatal assessment on the prognosis of congenital diaphragmatic hernia.
After reading in depth the manuscript, I would like to make some comments and ask the authors several questions about.

- Revise the document so that the sentences end properly. For example in the abstract, in the first line or in introduction section, page 2, line 52.

- There are few references in the introduction. A few more should be incorporated in order to be conducive to the objectives of the work.

- In page 3, line 113: “Recently, it has been demonstrated that the point at which CDH is diagnosed correlates with prognosis: early diagnosis is associated with a higher mortality rate [13].” I don't understand this sentence. Early diagnosis is supposed to be better, isn't it?

Round 2

Reviewer 1 Report

All issues were answered appropriately. This manuscript is a very nice addition to the current literature.